# Identification of Potent Zika Virus NS5 RNA-Dependent RNA Polymerase Inhibitors Combining Virtual Screening and Biological Assays

**DOI:** 10.3390/ijms24031900

**Published:** 2023-01-18

**Authors:** Ying Chen, Xiangyin Chi, Hongjuan Zhang, Yu Zhang, Luyao Qiao, Jinwen Ding, Yanxing Han, Yuan Lin, Jiandong Jiang

**Affiliations:** State Key Laboratory of Bioactive Substance and Function of Natural Medicines, Institute of Materia Medica, Chinese Academy of Medical Sciences and Peking Union Medical College, Beijing 100050, China

**Keywords:** Zika virus, RdRp, virtual screening, inhibitor, posaconazole

## Abstract

The Zika virus (ZIKV) epidemic poses a significant threat to human health globally. Thus, there is an urgent need for developing effective anti-ZIKV agents. ZIKV non-structural protein 5 RNA-dependent RNA polymerase (RdRp), a viral enzyme for viral replication, has been considered an attractive drug target. In this work, we screened an anti-infection compound library and a natural product library by virtual screening to identify potential candidates targeting RdRp. Then, five selected candidates were further applied for RdRp enzymatic analysis, cytotoxicity, and binding examination by SPR. Finally, posaconazole (POS) was confirmed to effectively inhibit both RdRp activity with an IC_50_ of 4.29 μM and the ZIKV replication with an EC_50_ of 0.59 μM. Moreover, POS was shown to reduce RdRp activity by binding with the key amino acid D666 through molecular docking and site-directed mutation analysis. For the first time, our work found that POS could inhibit ZIKV replication with a stronger inhibitory activity than chloroquine. This work also demonstrated fast anti-ZIKV screening for inhibitors of RdRp and provided POS as a potential anti-ZIKV agent.

## 1. Introduction

Zika virus (ZIKV) is a member of the *Flaviviridae* family, transmitting through mosquitoes, sex intercourse or vertical transmission from mother to fetus [1]. ZIKV infection during pregnancy can lead to congenital malformation, microcephaly and fetal demise. In adult patients, other neuroinflammatory diseases such as Guillain–Barré syndrome have been reported [2]. The ZIKV pandemic spread over 48 countries and territories in the Americas between 2015 and 2016, infecting more than 700,000 people [3]. Concerning a global health crisis, WHO declared ZIKV a public health emergency of international concern in February, 2016 [4]. Although ZIKV case numbers have decreased worldwide in recent years, it still has the potential to became a pandemic. However, there are currently no vaccines or drugs approved for the prevention or treatment of ZIKV infection [5].

The ZIKV genome is a single-stranded positive-sense RNA, encoding a polyprotein of three structural proteins (E, C and prM) and seven non-structural proteins (NS1, NS2A, NS2B, NS3, NS4A, NS4B and NS5) [6,7]. NS5 is the largest non-structural protein which has a RNA-dependent RNA polymerase (RdRp) domain at the C-terminal and a methyltransferase (MTase) domain at the N-terminal. RdRp initiates and governs the elongation of the RNA strand that includes the addition of nucleotides [8]. Since NS5 RdRp plays an important role in the replication and transcription of ZIKV and is absent in human beings, it has been proposed as a potential drug target for anti-ZIKV agents [9].

RdRp, a commonly conserved component of RNA viruses, is widely used as a target for antiviral drug screening [10]. Sofosbuvir is approved for the treatment of hepatitis C virus infection, primarily targeting RdRp. It is a nucleoside analog inhibitor, which is converted into its phosphorylated form in the liver to compete with the natural substrates of RNA synthesis at the active site of RdRp [11,12]. Azvudine (FNC) is also a nucleoside analog with anti-HIV activity, which can be intracellularly converted into azvudine triphosphate and inhibit HIV-1 RdRp activity [13]. It has been investigated for use against AIDS. Recently, azvudine showed therapeutic potential for the treatment of patients infected with SARS-CoV-2. It was conditionally approved to treat adult patients with COVID-19 in China [14]. Therefore, screening for inhibitors targeting RdRp can effectively accelerate the development of anti-ZKIV agents.

In this study, we performed a virtual screening of 935 compounds from an anti-infection compound library and a natural product library targeting ZIKV NS5 RdRp. Combined with a series of biological assays, we finally identified that posaconazole (POS) inhibited the activity of NS5 RdRp with an IC_50_ of 4.29 μM by binding with the key amino acid D666. Importantly, POS exhibited anti-ZIKV activity with an EC_50_ of 0.59 μM, more effective than chloroquine.

## 2. Results

### 2.1. Virtual Screening of NS5 RdRp Inhibitors

To screen for potential ZIKV NS5 RdRp inhibitors, virtual screening and biological assays were performed. The flow chart of the study designed is depicted in Figure 1.

Discovery Studio 2018R2 software is a powerful software for computer-aided drug design. LibDock program was employed for rapid docking, which calculated hotspots using grids at the binding sites and polar and non-polar probes [15]. In this research, virtual screening was performed using the LibDock protocol. As the LibDock score was the comprehensive evaluation for the docked poses, the compounds were ranked by the LibDock score [16]. ATP, the natural substrate of RdRp, was chosen as the reference compound. Then, the database of 935 compounds (MedChem Express, MCE, Monmouth Junction, NJ, USA) were docked into the active site of the RdRp. After screening, five compounds (DAB, GOS, POS, ITR and SUL) were found to have higher LibDock scores than ATP (LibDock score: 146.38) and were selected for further research. The virtual screening information of the top five compounds are listed in Table 1.

### 2.2. Biological Assays of NS5 RdRp Inhibitors

#### 2.2.1. The Inhibitory Effect of Compounds on NS5 RdRp Activity

As the compounds could be docked into the active site of the RdRp, they might inhibit the activity of RdRp. First, we purified a 6× His-tagged RdRp, verified by SDS-PAGE and Western blotting (Appendix A). The activity of RdRp was measured by a fluorescence-based alkaline phosphatase-coupled polymerase assay established previously in our lab [17]. Briefly, RdRp catalyzed the transfer of adenosine 5′-monophosphate from γ-(BBT)-ATP to the RNA chain, generating the byproduct BBTppi. Subsequently, BBTppi was hydrolyzed by alkaline phosphatase to fluorescent BBT (ex/em 430/560 nm) [18]. As was shown in Appendix A, the fluorescence intensity of the RdRp reaction gradually increased with time (0–70 min), indicating strong RdRp activity.

These selected five compounds were incubated with RdRp for 30 min before the enzymatic activity was examined. Different concentrations of DAB (1.25–80 μM), GOS (1.25–80 μM), POS (1.25–40 μM), ITR (0.625–80 μM) and SUL (2.5–80 μM) were applied. It was shown that all five compounds exhibited an inhibitory effect on RdRp activity in a dose-dependent manner. The IC_50_s of DAB, GOS, POS, ITR and SUL to inhibit RdRp activity were 11.28, 2.54, 4.29, 1.82 and 15.47 μM, respectively (Figure 2A). GOS, POS and ITR exhibited a more potent inhibitory effect on RdRp than DAB and SUL, with IC_50_s < 5 μM. Finally, we selected GOS, POS and ITR for further study.

#### 2.2.2. Cytotoxicity of Compounds

To evaluate the cytotoxicity of the selected compounds, different concentrations of ITR (1.25–200 μM), GOS (1.5625–40 μM) and POS (1.25–200 μM) were added to Huh-7 cells to conduct the WST-8 assay. As illustrated in Figure 2B, no cell toxicity was observed in cells treated with ITR. POS exhibited cytotoxicity with a CC_50_ of 101 μM. GOS could induce obvious cell toxicity at a concentration of 5 μM (CC_50_ = 7.79 μM). Although GOS showed a potent inhibitory effect on RdRp activity (IC_50_ = 2.54 μM), it had obvious cytotoxicity. Therefore, ITR and POS were selected for further experiments.

#### 2.2.3. Interaction of Compounds with NS5 RdRp by SPR

SPR can provide real-time observations of biomolecular interactions. It has been widely used to detect the direct interaction between proteins and small molecules [19]. As both ITR and POS had an obvious inhibitory effect against RdRp activity, the interaction of ITR and POS with RdRp was investigated. Thus, SPR assays were carried out. RdRp protein was immobilized on a CM5 sensor chip through amine coupling. The response units (RUs) of POS with RdRp showed a dose-dependent increase at concentrations of 3.125–50 μM POS (Figure 2C). The equilibrium dissociation constant (*K*_D_) was 2.76 × 10^−5^ M. However, there was almost no binding signals between ITR (3.125–50 μM) and RdRp (Figure 2C). Thus, we speculated that the inhibitory effect of POS on RdRp may be due to the direct binding to RdRp, while the RdRp inhibition mechanism by ITR might be different. POS was finally selected as the candidate for the mechanism study.

### 2.3. The Mechanism of POS against NS5 RdRp

#### 2.3.1. Molecular Docking of POS and NS5 RdRp

To further explore the inhibitory mechanism of POS on RdRp, molecular docking was performed using Discovery Studio 2018R2 software by the CDOCKER program. The docking model showed that POS occupied the activity center of RdRp (Figure 3A). The conventional hydrogen bond interaction was formed between a hydrogen atom of POS and an oxygen atom of D666. Pi–anion interaction was formed between the triazolone ring of POS and an oxygen atom of D666. Furthermore, a nitrogen atom of POS can interact with a hydrogen atom of K691, where carbon hydrogen bond was observed (Figure 3B). Since conventional hydrogen bonding interactions represented the major stabilizing force, D666 was speculated to be the key amino acid for the interaction between POS and RdRp.

#### 2.3.2. The Inhibitory Effect of POS on NS5 RdRp Mutants

The molecular docking results show that D666 plays a critical role in the binding of POS and RdRp. To further determine the inhibitory mode, D666 was mutated to A666. As a control, K470, located out of the activity center, was mutated to A470. The mutant proteins were purified and identified by SDS-PAGE and Western blotting, which were defined as D666A-RdRp and K470A-RdRp, respectively (Appendix A). Both D666A-RdRp and K470A-RdRp exhibited catalytic activities, with D666A-RdRp slightly less (Appendix A). Then, the inhibitory effect of POS on D666A-RdRp and K470A-RdRp was investigated. As expected, the IC_50_ of POS on K470A-RdRp was 3.64 μM, consistent with that on RdRp (4.29 μM). In contrast, the IC_50_ of POS on D666A-RdRp was 17.00 μM, approximately 4-fold higher than that of RdRp (Figure 4A). Therefore, we inferred that D666 was the key amino acid in the interaction between POS and RdRp.

#### 2.3.3. Interaction Detected by SPR between POS and D666A-RdRp

To further investigate the direct interaction of POS with RdRp, an SPR assay was performed. D666A-RdRp was immobilized on a CM5 chip surface. Interestingly, the RUs for the binding of POS with D666A-RdRp were obviously lower than that with RdRp. The *K*_D_ of POS on D666A-RdRp was 2.49 × 10^−4^ M, almost 10-fold higher than that on RdRp (2.76 × 10^−5^ M), indicating a weaker binding interaction between POS and D666A-RdRp (Figure 4B). These data demonstrate that D666 is likely to play an essential role in the binding of POS and D666A-RdRp. Therefore, the inhibitory effect of POS may be caused by its direct binding with RdRp via the key amino acid D666.

### 2.4. Anti-ZIKV Activity of POS

The anti-ZIKV activity of POS was determined by FACS. Huh-7 cells were infected with ZIKV (SMGC-1) and incubated with different concentrations of compounds. After 48 h, the cells were fixed and permeabilized, followed by staining with the FITC-anti-ZIKV E protein antibody. When detected by FACS, the FITC intensity in the given cell population reflected the ZIKV E protein-positive cells, indicting the cells infected with ZIKV. Chloroquine, a commonly used anti-malarial drug, was used as a positive control, as it has been demonstrated to inhibit ZIKV infection in vitro and in vivo [20,21,22]. Figure 5A shows that POS and chloroquine significantly reduced the number of ZIKV E protein-positive cells in a dose-dependent manner. The EC_50_s of POS and chloroquine were 0.59 μM and 2.80 μM, respectively, suggesting a potent anti-ZIKV activity of POS. Previous studies have shown that chloroquine exhibits no significant cytotoxicity to Huh-7 cells ≤ 10 μM [23]. Figure 5B illustrates that the cytotoxicity of POS at the indicated concentrations (≤ 6.4 μM) used in the anti-ZIKV activity was low, comparable to that of chloroquine. These results suggest that POS can inhibit ZIKV infection in Huh-7 cells more effective than chloroquine.

## 3. Discussion

Virtual screening is a promising in silico technique for drug design, which is based on algorithms and computational models to select compounds from a large compound database that are more likely to bind to specific biological targets [24]. Compared with the traditional drug screening process, virtual screening can greatly reduce the costs and increase hit rates to accelerate the drug development process [25]. Therefore, virtual screening has been widely used in drug discovery. There have been some successful applications in identifying bioactive leads. Kumarasiri M et al. reported by virtual screening that seven drug-like molecules, identified as potential cyclin-dependent kinase (CDK) 8 inhibitors, were active against colorectal cancer cell lines [26]. Another virtual screening model was established to successfully find acetylcholinesterase (AChE) inhibitors with the ability of protecting human neuroblastoma cells from Aβ-induced injury for Alzheimer’s disease treatment [27]. Furthermore, in the onset of the COVID-19 pandemic, there was a strong interest in rapidly and economically finding effective anti-coronavirus agents. Scientists performed virtual screening towards drug targets including the spike protein, 3CL protease and RdRp, identifying several potential anti-coronavirus compounds [28,29,30]. In the previous studies, virtual screenings against ZIKV proteins (RdRp, NS2B and NS3) were employed to find some small molecule compounds with anti-ZIKV activities [31,32,33]. However, most of these studies focused on the virtual screening alone, in which the mechanism studies of the active compounds were limited. To address these limitations, after virtually screening compounds against RdRp, we performed a series of biological assays to study the mechanism of the compounds. Therefore, this work provides a fast virtual screening and biological mechanism assays for anti-ZIKV agents.

RdRp, essential for viral RNA replication in host cells, is one of the most promising drug targets for RNA virus drug development. A robust RdRp activity assay is necessary for the rapid development of RdRp inhibitors. Current assays for RdRp activity include the cell-free biochemical RdRp enzyme activity assay and the cell-based assay [34]. In the cell-free system, RdRp activity acts on RNA elongation to produce quantitatively detectable substrates. Eltahla et al. reported a fluorometric assay to screen inhibitors of HCV RdRp, in which RdRp catalyzed the formation of double-stranded RNA that can be detected by PicoGreen [35]. A scintillation proximity assay was established to synthesize RNA containing radioactive nucleotides catalyzed by Dengue virus RdRp, leading to corresponding radioactive signals [36]. The fluorescence-based alkaline phosphatase-coupled polymerase assay was another biomedical assay, which incorporates a modified nucleotide analog in the substrate of RNA synthesis by RdRp, resulting in the release of a fluorophore for detection [18]. In this research, BBT-ATP was used to produce a highly fluorescent BBT to detect the activity of ZIKV RdRp. While cell-free RdRp enzyme activity assays have the advantage of initial high-throughput screening for RdRp inhibitors: however, some disadvantages have been recognized mainly for cellular metabolic effects of the compound [34]. To complement the cell-free assay, cell-based RdRp assays have been developed. In the cell-based assays, cells were transfected with RdRp together with the negative-sense luciferase RNA which can be transcribed into the positive-sense luciferase RNA by RdRp for reporter gene expression [37]. The inhibitors of RdRp for HCV, influenza and Lassa virus have been successfully identified using the cell-based assay [37,38,39]. However, most cell-based assays were on transiently transfected cell lines, suffering from unstable transfection efficiency and poor reproducibility [40]. Accordingly, the method for detecting RdRp activity would be instrumental in screening and validating the inhibitors of RdRp.

It has been reported that D535, D665 and D666 are crucial in the activity site of RdRp [41]. By molecular docking, D666 was shown to be the key amino acid for the interaction of POS and RdRp. When D666 was mutated to alanine, there was no significant difference regarding RdRp activity between the WT and D666A mutant. We speculated that the activity of RdRp was relatively little affected by one mutation out of three key amino acids. As expected, D666A-RdRp showed a clear resistance to POS (IC_50_ = 17.00 μM), approximately 4-fold higher than the wild-type. However, it seemed that D666A-RdRp was still sensitive to POS. We inferred that there might be other inhibitory mechanisms of POS on RdRp, which remained largely unexplored. The co-crystallization of RdRp and POS could help elucidate the exact inhibitory mechanism of POS in the future. In the previous study, G664, D665 and D666 on RdRp were all mutated to alanine in ZIKV replicon RNA, followed by electroporation into Huh-7 cells. Only a low level of viral RNA replication was detected, indicating G664, D665 and D666 were essential in viral replication [42]. Thus, the influence of the D666A-RdRp mutation into ZIKV replicon on viral replication and the effect of POS under the D666A genetic background were crucial for evaluating the activity of POS. In subsequent experiments we will further explore the activity and mechanism of POS.

POS is a triazole compound with various biological activities, such as antifungal, antitumor and antiviral. One of the most commonly used activities of POS is its antifungal properties. POS disrupted the biosynthesis of ergosterol, damaging plasma membrane function and ultimately leading to the death of fungal cells [43]. Concerning antitumor activity, POS exhibited an inhibitory effect on the growth of basal cell carcinoma through deregulating the hedgehog pathway which affects cell proliferation and differentiation [44]. POS also acted as an anti-breast cancer agent by inhibiting cytochrome P450 27A1 activity [45]. Furthermore, POS has been shown to possess antiviral activities against alphaviruses, human cytomegalovirus and dengue virus [46,47,48]. It was reported that POS inhibited alphavirus replication by modulating cholesterol trafficking and slowing clathrin-mediated endocytosis [46]. For the anti-human cytomegalovirus activity, POS inhibited the human CYP51 protein which plays an essential role in host cholesterol biosynthesis and the maintenance of sterol homeostasis [47]. Scientists also found that POS reduced the replication of another flavivirus dengue virus in a dose-dependent manner with an EC_50_ of 4.1 μM by targeting the oxysterol-binding protein (OSBP), which mediates the altered intracellular cholesterol distribution [48]. In a previous study, POS was shown to inhibit the replication of ZIKV sub-genomic replicon RNA infected in Hela cells at a single concentration of 10 μM [48]. However, the distinct mechanism and anti-ZIKV activity of POS are unknown. In our study, we confirmed that POS has a direct inhibitory effect on ZIKV RdRp and exhibited anti-ZIKV activity with an EC_50_ of 0.59 μM.

In summary, combining virtual screening and biological approaches, POS has been identified as the inhibitor of ZIKV NS5 RdRp by binding with D666 of the active site. Moreover, POS exhibited the anti-ZIKV activity superior to chloroquine. Thus, POS has the potential to be a lead compound. In future, we hope to optimize the anti-ZIKV activity and toxicity of POS by structural modification. The results and methods in this study have also provided some insights into COVID-19 drug discovery.

## 4. Materials and Methods

### 4.1. Cells and Virus

Human hepatocellular carcinoma Huh-7 cells were obtained from the Cell Resource Center, Peking Union Medical College (National Infrastructure of Cell Line Resource, NSTI, Beijing, China). The cell line was maintained at 37 °C and 5% CO_2_ in Dulbecco’s Modified Eagle Medium (DMEM, Gibco, Grand Island, NY, USA), which contained 10% fetal bovine serum (FBS, Gibco, Grand Island, NY, USA) and 1% penicillin/streptomycin (Gibco, Grand Island, NY, USA). ZIKV (SMGC-1) was provided by Institute of Microbiology, Chinese Academy of Sciences. Cells treated with virus were fixed and permeabilized using Fixation/Permeabilization Solution Kit (BD Bioscience, San Diego, CA, USA). Virus titer was determined by fluorescence-activated cell sorting (FACS, BD Bioscience, San Diego, CA, USA).

### 4.2. Chemicals and Antibodies

The database of 935 compounds, comprising an anti-infection compound library and a natural product library, was obtained from MedChem Express (MCE, Monmouth Junction, NJ, USA) for virtual screening. All the compounds used in this study were purchased from MedChem Express (MCE, Monmouth Junction, NJ, USA). Anti-6× His Tag antibody was purchased from Abcam (1:1000 diluted), and anti-rabbit IgG HRP-linked antibody was purchased from Cell Signaling Technology (1:5000 diluted). FITC-anti-ZIKV E protein antibody was provided by Institute of Microbiology, Chinese Academy of Sciences.

### 4.3. Virtual Screening

Discovery Studio 2018R2 software (Accelrys, San Diego, CA, USA) was applied to perform the virtual screening. The crystal structure of ZIKV NS5 RdRp with a resolution of 1.90 Å was downloaded from the Protein Data Bank (PDB ID: 5U04). RdRp protein was prepared to remove the heteroatoms and water, add hydrogen, protonate, ionize and minimize energy. 935 compounds from an anti-infection compound library and a natural product library were all prepared and different conformations of the compounds were optimized. ATP, the natural substrate of RdRp, was used as a control. The docking pocket of RdRp was defined according to the key amino acids of D535, D665 and D666 reported previously [41]. The virtual screening of NS5 RdRp inhibitors was carried out by docking all the prepared ligands at the defined docking pocket using the LibDock program. Based on the LibDock score, all the docking poses were ranked. Compounds with the top five LibDock scores were selected for further study.

### 4.4. NS5 RdRp Enzymatic Inhibition Assay

The plasmid pET30a-RdRp, constructed in our laboratory, was transformed into *E. coli* Transetta (DE3) for RdRp expression and purified [17]. The bacteria were grown in ZYM-5052 culture medium. RdRp with the His tag was loaded on a Ni^2+^ HisTrap chelating column and then eluted in a buffer containing imidazole in the ÄKTA system. The purified proteins were verified by SDS-PAGE and Western blotting. RdRp activity was analyzed using a fluorescence-based alkaline phosphatase-coupled polymerase assay. The inhibitory effect of the compounds against RdRp was evaluated using the protocol developed in our laboratory [17].

### 4.5. Cell Viability Assay

The WST-8 assay was used to evaluate the cytotoxicity of the compounds. Huh-7 cells (1 × 10^4^ cells/well) were seeded in a 96-well plate and different concentrations of compounds were added the next day. After a 24 h incubation, the supernatant was discarded and WST-8 reagent (1:10 diluted by DMEM) was added. The cell viability was assessed using a Microplate Reader (ELx808, Biotek, North Chesterfield, VA, USA) at 450 nm.

### 4.6. Surface Plasmon Resonance (SPR) Assay

The SPR was carried out using BIAcore T200 (GE Healthcare, Uppsala, Sweden) at 25 °C. RdRp was diluted by 10 mM sodium acetate (pH 4.5) and then immobilized on a CM5 sensor chip using an NHS/EDC amine-coupling kit in 1× PBS-P running buffer. Different concentrations (3.125–50 μM) of compounds flowed through the sensor chip surface and the RUs was detected. The equilibrium dissociation constant (*K*_D_) was fitted using the BIAcore Evaluation Software.

### 4.7. Molecular Docking

The binding mode between POS and RdRp was further analyzed by the Discovery Studio 2018R2 software using the CDOCKER program. The docking pocket was defined based on the key amino acids of D535, D665 and D666. The docking mode and the key amino acids between POS and RdRp were determined by energy scores and binding types in the receptor–ligand interactions. To verify the key amino acids, we mutated D666 to A666 using the Fast Mutagenesis System (TransGen Biotech, Beijing, China). In addition, K470, an amino acid outside the active site, was also mutated to A470 as a control. They were designated as D666A-RdRp and K470A-RdRp, respectively.

### 4.8. Anti-ZIKV Activity

Huh-7 cells were seeded into 24-well plates and infected with ZIKV (SMGC-1) at a multiplicity of infection (MOI) of 0.2. Then, different concentrations of POS (0.05–6.25 μM) diluted in culture medium were added and incubated for 48 h. Chloroquine was used as a positive control. Cells were fixed and permeabilized for 20 min using a Fixation/Permeabilization Solution Kit. After washing, the cells were incubated with FITC-anti-ZIKV E protein antibody for 30 min. Then, ZIKV E protein-positive cells were detected in a FACS Analyzer and analyzed by FlowJo (TreeStar, Ashland, OR, USA). The EC_50_s of the compounds were calculated as the reduction ratio of ZIKV E protein-positive cells over the concentrations of the compounds. Meanwhile, the cytotoxicity of POS in Huh-7 cells was evaluated by WST-8 assay as described before.

### 4.9. Statistical Analysis

The IC_50_, CC_50_ and EC_50_ values were all calculated using the GraphPad Prism 8 software.

## Figures and Tables

**Figure 1 ijms-24-01900-f001:**
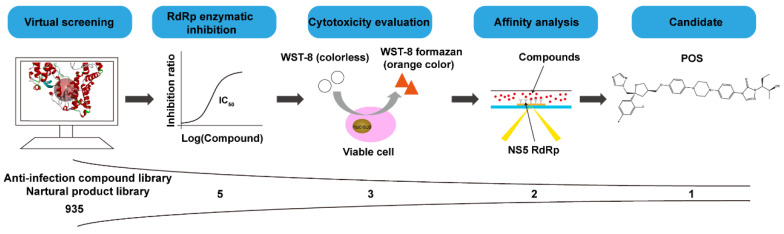
Flow chart depicting the methodology used in the screening of ZIKV NS5 RdRp inhibitors. 935 compounds from an anti-infection compound library and a natural product library were applied to the virtual screening by Discovery Studio in LibDock program. Five compounds, with the top LibDock scores, were chosen to determine the inhibitory effects on RdRp. Three compounds, which had higher inhibitory effects on RdRp, were applied to the cytotoxicity analysis by WST-8. Excluding a compound that was toxic to cells, two compounds were chosen to examine the affinities with RdRp by SPR. At last, POS was selected as the candidate.

**Figure 2 ijms-24-01900-f002:**
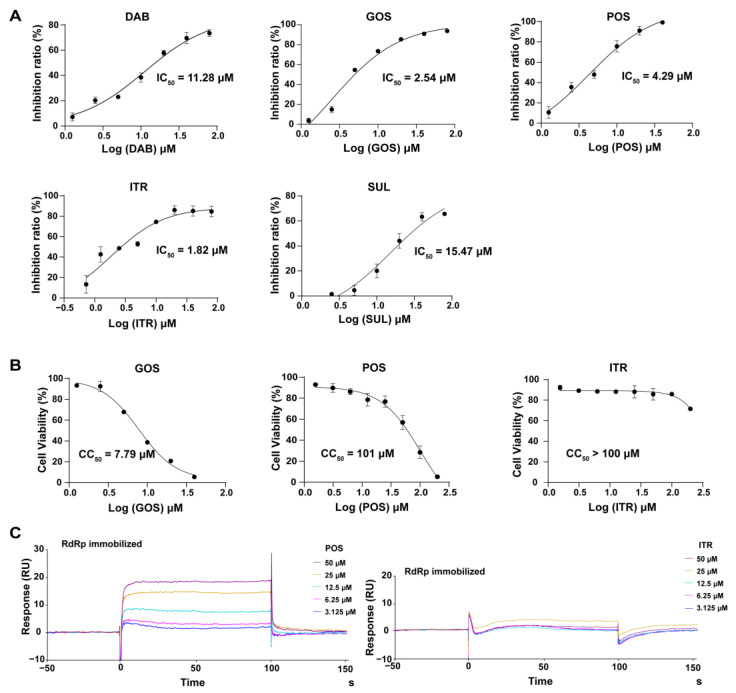
Biological assays of RdRp inhibitors. (**A**) The inhibitory effect of different compounds on RdRp activity. Different concentrations of DAB (1.25–80 μM), GOS (1.25–80 μM), POS (1.25–40 μM), ITR (0.625–80 μM) and SUL (2.5–80 μM) were applied. The IC_50_s were calculated as the inhibition ratios of the fluorescence intensity over the concentration of the compound. DAB, GOS, POS, ITR and SUL inhibited the activity of RdRp with IC_50_s of 11.28, 2.54, 4.29, 1.82 and 15.47 μM, respectively. (**B**) The cytotoxicity of the compounds on Huh-7 cells. Different concentrations of ITR (0–200 μM), GOS (0–40 μM) and POS (0–200 μM) were applied. Cell viability was detected by a WST-8 assay at 450 nm. The CC_50_s of GOS, POS and ITR were 7.79, 101 and >100 μM, respectively. (**C**) The binding of compounds with RdRp by SPR. RdRp was immobilized on a CM5 sensor chip. Different concentrations of POS (3.125–50 μM) and ITR (3.125–50 μM) flowed over the RdRp surface, resulting in real-time changes in RU.

**Figure 3 ijms-24-01900-f003:**
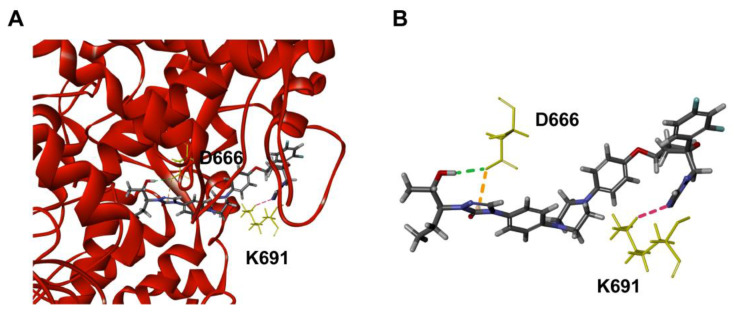
Molecular docking of RdRp and POS. (**A**) The overview of POS bound with the docking pocket of RdRp. The yellow sticks represent the amino acids interacting with POS. The stick model represents POS (light gray, hydrogen atoms; deep gray, carbon atoms; red, oxygen atom; purple, nitrogen atoms; and cyan, fluorine atom). (**B**) The detailed intermolecular bonds between POS and RdRp. Conventional hydrogen bond between D666 and POS is shown in green. Pi–anion interaction is orange. Pink bond represents the carbon hydrogen bond interaction between K691 and POS.

**Figure 4 ijms-24-01900-f004:**
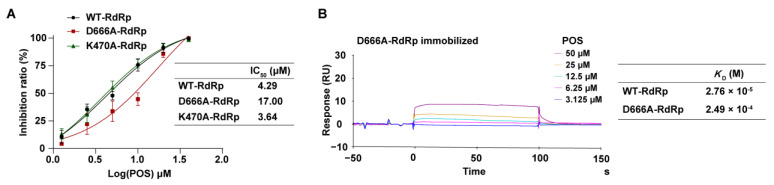
The mechanism of the inhibitory effect for POS. (**A**) The inhibitory effect of POS (1.25–40 μM) on RdRp, D666A-RdRp and K470A-RdRp. IC_50_s are plotted as the inhibition ratio of the fluorescence intensity over the concentration of compounds. The IC_50_s of POS for WT-RdRp, D666A-RdRp and K470A-RdRp were 4.29 μM, 17.00 μM and 3.64 μM, respectively. (**B**) The affinity analysis between POS (3.125–50 μM) and D666A-RdRp by SPR. The *K*_D_ of POS on D666A-RdRp was 2.49 × 10^−4^ M, which was 10-fold higher than that on WT-RdRp (2.76 × 10^−5^ M).

**Figure 5 ijms-24-01900-f005:**
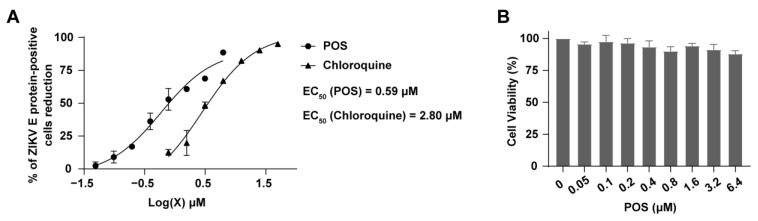
The anti-ZIKV activity of POS. (**A**) Anti-ZIKV activity detected by FACS. Huh-7 cells were infected by ZIKV (SMGC-1) and incubated for 48 h with POS (0.05–6.25 μM) or chloroquine (0.78–50 μM). Cells were stained with FITC-anti-ZIKV E protein antibody and analyzed by FACS. Results are expressed as the percentage reduction in the number of ZIKV-infected cells after being treated with compounds. The EC_50_s were calculated as the reduction ratio of ZIKV E protein-positive cells over the concentration of POS. The EC_50_s of POS and chloroquine were 0.59 μM and 2.80 μM, respectively. (**B**) The cytotoxicity of POS evaluated by WST-8 assay. The viability of Huh-7 cells were examined at 24 h after the addition of POS at the indicated concentrations (0–6.4 μM).

**Table 1 ijms-24-01900-t001:** Top five ranked compounds with LibDock scores.

No.	Compound Name	Structure	LibDock Score	Absolute Energy	Relative Energy	Conf Number
1	Dabigatran etexilate (DAB)	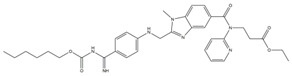	164.77	99.27	6.00	10
2	Gossypol (acetic acid) (GOS)	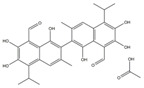	158.62	93.92	19.20	40
3	Posaconazole (POS)	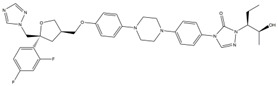	147.89	115.52	16.73	137
4	Itraconazole (ITR)	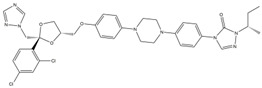	147.33	117.48	14.03	131
5	Sulconazole (nitrate) (SUL)	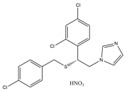	146.89	55.28	8.26	28

## Data Availability

The original data are available upon request from the authors.

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
