# Peer review of "Identification of Potent Zika Virus NS5 RNA-Dependent RNA Polymerase Inhibitors Combining Virtual Screening and Biological Assays"

_ijms, 2023, doi:10.3390/ijms24031900_

Round 1

Reviewer 1 Report

in this manuscript, the authors studied the discovery of potential anti-ZIKA virus through the employment of virtual screening and biological studies.

the rational and approach is sufficient and well-explained. The support of biological data granted the study additional supportive to the virtual screening. 

in line 79, the authors should mention what is the name of the database used (i.e., in-house, downloaded from...?)

Author Response

Response to Reviewer 1 Comments

In this manuscript, the authors studied the discovery of potential anti-ZIKA virus through the employment of virtual screening and biological studies.

The rational and approach is sufficient and well-explained. The support of biological data granted the study additional supportive to the virtual screening.

In line 79, the authors should mention what is the name of the database used (i.e., in-house, downloaded from...?)

Response: Thank you for your suggestions. The compound database, comprised of an anti-infection compound library and a natural product library, was obtained from MedChem Express (MCE, USA). This has been stated in Results Section (Page 2, Line 79-80) and Materials and Methods Section (Page 8, Line 308-310).

Reviewer 2 Report

Zika virus  epidemic poses a significant threat to human and Since until now there is no drug against it, I think this article is interesting, because it proposes such a drug. Authors performed a virtual screening of 935 compounds from an antiinfection compound library and a natural product library. In the introduction, the authors briefly but clearly present the current stage in the treatment of this type of virus and explain the reasoning behind it. Compounds from an anti-infection compound Library and a natural product library were applied to the virtual screening by Discovery Studio in LibDock program. 5 compounds were chosen to determine the inhibitory effects on RdRp. 3 compounds, which had higher inhibitory effects on RdRp, were applied to the cytotoxicity analysis by WST- 8. Excluding a compound that was toxic to cells, 2 compounds were chosen to examine the affinities with RdRp by SPR. At last, one was selected as the candidate. Each step of this methodology is well presented and argued.Posaconazole was found to be a potential drug for this disease caused by Zika. Materials and methods chapter is well written and clear. The bibliography is recent and uniformly written.

Overall, the article deserves to be published, but it needs additions. The authors did not present any clear conclusions from which to derive what are the most important aspects of this study and what or if they propose any compound as drug.

Author Response

Response to Reviewer 2 Comments

Zika virus epidemic poses a significant threat to human and Since until now there is no drug against it, I think this article is interesting, because it proposes such a drug. Authors performed a virtual screening of 935 compounds from an antiinfection compound library and a natural product library. In the introduction, the authors briefly but clearly present the current stage in the treatment of this type of virus and explain the reasoning behind it. Compounds from an anti-infection compound Library and a natural product library were applied to the virtual screening by Discovery Studio in LibDock program. 5 compounds were chosen to determine the inhibitory effects on RdRp. 3 compounds, which had higher inhibitory effects on RdRp, were applied to the cytotoxicity analysis by WST-8. Excluding a compound that was toxic to cells, 2 compounds were chosen to examine the affinities with RdRp by SPR. At last, one was selected as the candidate. Each step of this methodology is well presented and argued. Posaconazole was found to be a potential drug for this disease caused by Zika. Materials and methods chapter is well written and clear. The bibliography is recent and uniformly written.

Overall, the article deserves to be published, but it needs additions. The authors did not present any clear conclusions from which to derive what are the most important aspects of this study and what or if they propose any compound as drug.

Response: Thank you for your constructive suggestions. In this study, we screened inhibitors of ZIKV NS5 RdRp combining virtual screening and biological assays. Finally, POS was identified to inhibit ZIKV replication with a stronger inhibitory activity than chloroquine and had the potential to be the lead compound against ZIKV infections. Besides, the results and methods in this study have also provided some insights into the COVID-19 drug discovery. These were stated in Abstract Section (Page 1, Line 20-23) and Discussion Section (Page 8, Line 290-295).